# An Integrated Multi-Channel Deep Neural Network for Mesial Temporal Lobe Epilepsy Identification Using Multi-Modal Medical Data

**DOI:** 10.3390/bioengineering10101234

**Published:** 2023-10-21

**Authors:** Ruowei Qu, Xuan Ji, Shifeng Wang, Zhaonan Wang, Le Wang, Xinsheng Yang, Shaoya Yin, Junhua Gu, Alan Wang, Guizhi Xu

**Affiliations:** 1State Key Laboratory of Reliability and Intelligence of Electrical Equipment, Hebei University of Technology, Tianjin 300401, China; 2021904@hebut.edu.cn (R.Q.); jixuan20212021@163.com (X.J.); 17860503045@163.com (Z.W.); xsyang@hebut.edu.cn (X.Y.);; 2Tianjin Universal Medical Imaging Diagnostic Center, Tianjin 300110, China; prescription@163.com; 3Department of Functional Neurosurgery, Huanhu Hospital, Tianjin 300350, China; wangledr@126.com (L.W.); yinsya@hotmail.com (S.Y.); 4Auckland Bioengineering Institute, The University of Auckland, Grafton, Auckland 1010, New Zealand; alan.wang@auckland.ac.nz

**Keywords:** mesial temporal lobe epilepsy, multi-modal data confusion, neural network, deep learning, epilepsy, medical image analysis

## Abstract

Epilepsy is a chronic brain disease with recurrent seizures. Mesial temporal lobe epilepsy (MTLE) is the most common pathological cause of epilepsy. With the development of computer-aided diagnosis technology, there are many auxiliary diagnostic approaches based on deep learning algorithms. However, the causes of epilepsy are complex, and distinguishing different types of epilepsy accurately is challenging with a single mode of examination. In this study, our aim is to assess the combination of multi-modal epilepsy medical information from structural MRI, PET image, typical clinical symptoms and personal demographic and cognitive data (PDC) by adopting a multi-channel 3D deep convolutional neural network and pre-training PET images. The results show better diagnosis accuracy than using one single type of medical data alone. These findings reveal the potential of a deep neural network in multi-modal medical data fusion.

## 1. Introduction

Epilepsy is a chronic brain disease with recurrent seizures and a prevalence of 1% [1]. It is one of the most serious chronic dysfunction syndromes caused by the abnormally synchronous discharge of neurons [2]. Epileptic seizures have the characteristics of being recurrent and sudden. They are accompanied by clinical manifestations such as convulsions of limbs, fainting and loss of consciousness. When a seizure occurs, the patient’s limbs are not controlled by autonomous consciousness, and even a brief seizure will bring physical and psychological damage to the patient. Severe seizures can lead to brain-cell death and can even be life-threatening. About 35% of seizures cannot be restrained using anticonvulsant drugs, which is called medically refractory epilepsy, which requires the removal of the epileptic zone (EZ) with radiofrequency ablation or surgery. Among all kinds of epilepsy, medial temporal epilepsy (MTLE), especially hippocampal sclerosis (HS), accounts for 65% of refractory focal epilepsy [3]. Removing the epileptic zone (EZ) with an operation is an important method for preventing epilepsy. It is generally applicable to a situation where the epileptogenic focus is clearly located. Accurate localization of the EZ is difficult and usually requires synthetic analysis and judgment from clinical symptoms, personal demographic and cognitive data (PDC), electroencephalography (EEG), positron emission tomography (PET), magnetic resonance imaging (MRI) and implanted electrodes. Therefore, the accurate diagnosis of epilepsy is extremely complicated [4].

Symptoms of epilepsy vary because the start positions of the discharge are different [5]. For example, seizures caused by MTLE-HS usually begin with an upper-abdominal pain and bloating. Then, the symptoms progress to a feeling of fear, culminating in generalized tonic–clonic seizures. This is because the feeling of fear is highly associated with the amygdala in the medial temporal lobe. For that reason, a feeling of fear can be used as one of the key characteristics of epilepsy classification and MTLE diagnostics. In various medical imaging examinations, MRI is a common diagnostic method for MTLE used to find structural abnormalities. Because of recurrent seizures, a pathological manifestation of HS is often the loss of hippocampal cells. This pathology shows decreased volume and increased density in the hippocampal region in a structural MRI and reduced metabolism in PET images. However, some MTLE shows no obvious structural abnormality in structural MRI, which is called structural-MRI-negative epilepsy, such as gliocytoma. In order to diagnosis MTLE more accurately, PET/CT is adopted to detect the metabolic alterations in EZ. PET/CT can effectively detect the lesions with no abnormal morphology and structure, but with changes in cell molecules. Since the epileptogenic EZ shows a hypo-metabolism during the interictal period, PET/CT is used to detect the functional and metabolic defects in the brain, thus helping to detect the minor lesions which are not detected using structural MRI.

In the latest research of epilepsy pre-surgery, some deep learning algorithms of single-modal and single-channel inputs have achieved good results. In the simple electroencephalogram (EEG) classification problems, classical machine learning performs well. Feature extraction and a support vector machine (SVM) were adopted for high-frequency oscillations (HFOs) detection in seizures of EEG to distinguish which channel has significant difference and may provide the surgery with effective guidance [6]. A patient-specific detector was proposed based on a micro-power SVM, which achieves a good result while consuming less power in seizure detection [7]. Then, System-on-Chip (SoC) was applied for patient-specific seizure detection to achieve better performance [8]. In medical image classification problems, a decision tree algorithm was adopted to identify two detailed types of epilepsy: childhood myoclonic and borderline absence syndrome [9]. A random forest algorithm was applied to the resting-state functional magnetic resonance imaging (MRI) [10], and a fast single shot proximal SVM was applied to functional MRI images to predict whether the epilepsy has a potential risk of seizure [11]. Five types of epilepsy were classified using SVM and structural MRI and showed a high classification performance of 91% [12]. Then, a cascade of machine learning classifiers were integrated, and bio-inspired heuristics were involved to detect the seizures [13].

A deep learning algorithm has been applied to the medical-image-processing field, especially in the area of epilepsy diagnosis. A neuro-fuzzy technique and artificial neural network (ANN) have been proposed to improve the efficiency and accuracy of epilepsy zone detection and diagnosis [14]. To reduce the computational complexity, a high-angle resolved diffusion imaging (HARDI) and neurite orientation dispersion and density imaging (NODDI) were adopted to improve the functional network. They are used to detect juvenile myoclonic epilepsy [15]. A multi-spike state learning was proposed to classify seizures and achieve an accuracy of 71.23% [16]. A transfer learning method was applied for further classification of the epileptic state and to improve the accuracy to 97% [17]. In order to study the spread path of abnormal discharge, a deep convolutional neural-network-tract classifier was adopted to distinguish the connectivity of white-matter fiber in children with focal epilepsy to improve the surgery effect [18]. In the recent research, a multi-modal medical-image-fusion approach was proposed to increase the epilepsy diagnosis accuracy [19].

With the development of medical technology, the diagnosis information can be obtained from different channels for the same disease. Presently, how to integrate multi-modal data for an automated diagnosis of MTLE is an urgent problem to be solved. Therefore, the object of this study is to propose a multi-channel deep network of integrated multi-modal inputs, including typical clinical symptoms, PDC, structural T1 MRI and PET/CT, and to evaluate the diagnosis performance between MTLE patients and a health control group by adopting a multi-channel 3D convolutional neural network.

## 2. Materials and Methods

### 2.1. Subjects

A total of 30 participants, 15 MTLE patients and 15 age-matched health controls, were continuously collected from Tianjin Universal Medical Imaging Diagnostic Center and Tianjin Huanhu Hospital. It is worth mentioning that these 15 health controls are patients who had a disease that required a PET/CT examination but were normal in the brain. This is because there is nuclear radiation in a PET/CT examination, and it is harmful to healthy people to undergo this experiment. All patients have received the same examinations of T1-MRI and PET/CT and ultimately have undergone surgery or thermocoagulation to deactivate their EZ. Exclusion criteria were as follows: (1) having generalized epilepsy, not focal epilepsy, and (2) having no EZ resection, so the clinical gold standards could not be obtained. The diagnoses and labels were made by a group of certified neurosurgeons and radiologists. The experiment was approved by the Biomedical Ethics Committee of Hebei University of Technology. All participants involved in this study have signed the informed-consent forms. Informed consent was obtained from all subjects involved in the study. All examinations were completed following the diagnostic protocols of the International League against Epilepsy (ILAE) [20].

### 2.2. MRI System and Image Pre-Processing

All MTLE patients and health controls were scanned within 11 min using a 3.0-T Ingenia MR scanner from Philips Healthcare. T1 images were obtained, followed by a turbo-spin echo sequence with a repetition time/inversion time/echo time of 6600/2000/330 ms. The acquisition parameters are as follows: number of coil channels is 32; flip angle is 180°; turbo-spin echo factor is 182; bandwidth is 1187.1 Hz; field-of-view is 220 × 180 × 150 mm^3^; matrix is 220 × 180 × 150; thickness of image is 1 mm; number of excitation is 2; number of slices is 150; CS-SENSE is 3; and spectral inversion recovery was applied for fat suppression.

The raw data of T1 MRI images of slices were recorded, originally in the format of Digital Imaging and Communications in Medicine (DICOM). In order to process the image data with the toolbox of statistical parametric mapping (SPM), the original MRI images were translated to the format of the Neuroimaging Informatics Technology Initiative (NIFTI). To register the realistic head image with the standardized Montreal Neurological Institute (MNI) template, six reference points are requested to be calibrated in the MRI images, which are the anterior commissure (AC), posterior commissure (PC), interhemispheric point (IH), nasion (NAS), left preauricular (LPA), and right preauricular (RPA). The anterior commissure (AC) is a bundle of nerve fibers located in front of the columns of the fornix, which connects the two cerebral hemispheres across the midline. The posterior commissure (PC) is a rounded band of white fibers crossing the middle line on the dorsal aspect of the upper end of the cerebral aqueduct. The interhemispheric point (IH) is at the top of the head and in the interhemispheric space. The nasion (NAS) is the intersection of the frontal and two nasal bones of the human skull, and is on the visible surface of the face between the eyes. The left preauricular (LPA) and right preauricular (RPA) are preauricular points located on either side of the head, which are very important to register bilateral cerebral hemispheres. Then, the cerebrum is divided in to 90 areas according to Anatomical Automatic Labeling (AAL) to improve the accuracy of the region of interests (ROI) extraction. This step is to better distinguish the functional and metabolic information alternations in PET images between different brain regions, and to obtain more accurate results of brain-region localization.

### 2.3. PET System and Image Pre-Processing

All participants fasted for at least 6 h before receiving the injection of ^18^F-FDG intravenously in an awake and resting state with their eyes closed. PET image scanning was started 40 min after the injection. The mean scanning time after the FDG injection was the same between the MTLE patients and healthy controls. The PET scanner was a Philips Vereos 64 HD PET/CT (Philips, The Netherlands). The parameters of scanning are as follows: the iteration is 2, number of iterations is 1, regularization is 6. The scanning of all participants lasted about 25 min (3.5 min/bed, 7~8 bed/subjects), and the ^18^F-FDG-PET image was reconstructed with an ordered-subset expectation maximization (OSEM) iterative reconstruction algorithm in a 168 × 168 matrix with a pixel size of 3.4 mm. Here, we adopted 3D Gaussion post filter to suppress high-frequency components to a greater extent, making the image smoother. In order to prevent the partial volume effect in PET images, point spread function (PSF) is used to restore the high-resolution image space efficiently, using the iterative Richardson–Lucy maximum likelihood algorithm.

The raw data of PET images of the slices were also recorded originally in the format of DICOM and translated to NIFTI. Then, they were registered to the structural MRI images, which were processed already to keep the same anatomical structure in the same location for subsequent analysis. The registered PET-MRI images were processed with SPM toolbox, followed by these five steps: (1) time slice and head motion calibration, (2) spatial normalization, (3) tissue segmentation, (4) 3D head-model reconstruction and (5) mapping to a Broadmann template. The 3D cerebral cortex after pre-processing, registration and mapping to a Broadmann template is shown in Figure 1. The Brodmann area is a system that divides the cerebral cortex into a series of anatomical regions based on cellular structure, which is also shown in Figure 1.

In PET images, the standard uptake value (SUV) is used as a quantitative indicator to assess tissue uptake of ^18^F-FDG tracer. In order to eliminate the differences between individuals, we used the average SUV of each subject’s cerebellar region as the standard to calculate the ratio of SUV (SUVR) of each brain region according to the AAL template.
(1)$${SUVR}_{i}=\frac{{SUVR}_{i}}{{SUVR}_{cerebellar}}$$
where *i* presents the brain regions from 1 to 90.

## 3. Experimental Design

### 3.1. Multi-Modal Data Integration

Considering that simple 2D convolutional kernels make it difficult to understand the spatial relationships of brain tissue adequately, we adopted 3D feature maps to construct the neural network architecture. After pre-processing, structural T1 MRI and PET images were divided into small cubes of 48 × 48 × 48 and input separately into 3D deep convolutional neural networks (CNN). The ROI matrices were calculated using the brain parcellation. In order to reduce the amount of computation, only the elements in the upper triangle of each ROI matrices were retained. In the other channel, a 1D vector was created to contain the clinical symptoms and PDC information, which was composed of the typical symptom, age, gender and years of education of each subject. The experimental design schematic is shown in Figure 2.

### 3.2. Multi-Channel Deep Neural Network

A multi-channel 3D CNN was built to classify between MTLE patients and health controls through TensorFlow, which, using various modals of data, included structural T1 MRI, ^18^F-FDG PET/CT, clinic symptom and PDC data. The architecture of the proposed network is shown in Figure 3.

This network consists of two main blocks. One is the image-data channel which consists of a 3D feature map that processes structural T1 MRI and ^18^F-FDG PET/CT images. The image-data channel contains two independent inputs, which run separately and then concatenate into one single output. The neural network in the first step is composed of an input layer, three 3D convolutional layers, three pooling layers and a dropout layer. After concatenation, the neural network is composed of two 3D convolutional layers, three pooling layers and a dropout layer. The other one is the clinical-information channel which consists of 1D feature map that processes typical clinical symptom and PDC data. The input of the clinical-information channel concatenated the feature vector of typical clinical symptom data and the feature vector of PDC data. The network related to this clinical-information channel consists of an input layer, two 1D convolutional layers, two pooling layers and a dropout layer. The outputs from these two channels were stretched and concatenated into one vector. The final integrated block after these two channels is composed of two convolutional layers, batch normalization and dropout layers. The number of channels corresponds to the number of feature maps in that layer. Finally, the output value was normalized to 0 or 1 to identify whether the input subject is an epilepsy patient or not.

A total of 10 patients and 10 healthy controls were divided into training set randomly and the others were the testing set. The training was run on a workstation with a M4000 NVIDIA GPU. The training parameters were as follows: epochs, 500; batch size, 1; loss function, binary cross-entropy; activation function, ReLU; learning rate: 0.1; and early stop, after 1000 iterations.

## 4. Results

### 4.1. Identification Result in Different Algorithms

A five-fold cross-validation was involved to eliminate bias. Accuracy, specificity, sensitivity, F1-measure and area under the curve (AUC) were calculated to evaluate the performance of the proposed multi-channel integrated neural network.
(2)$$Acc=\frac{\mathrm{TP}+\mathrm{TN}}{\mathrm{TP}+\mathrm{TN}+\mathrm{FP}+\mathrm{FN}},$$
(3)$$Spe=\frac{\mathrm{TP}}{\mathrm{TP}+\mathrm{FP}},$$
(4)$$Sen=\frac{\mathrm{TP}}{\mathrm{TP}+\mathrm{FN}},$$
(5)$$F1-measure=\frac{2•Sen•Spe}{Sen+Spe}$$
where TP, TN, FP and FN are the number of true positives, true negatives, false positives and false negatives, respectively. The MTLE classification result is shown in Table 1 and Figure 4.

As shown in Table 1 and Figure 4, the lowest accuracy and AUC were obtained when only using the PET image data and the image-data channel (Accuracy = 48.30% ± 19.85%, AUC = 0.50). The identification result was obtained only using the structural MRI data and the image-data channel (Accuracy = 54.45% ± 7.67%, AUC = 0.57). We speculate that structural anomalies can be detected more easily via the neural network. For this reason, when using both PET images and structural MRI images as the inputs of image-data neural network block, the accuracy and AUC arose to 67.72% ± 50.25% and 0.65. The highest accuracy and AUC were obtained when testing the whole integrated neural network using PET images, structural T1 MRI images, typical clinical symptom and PDC (Accuracy = 77.33% ± 14%, AUC = 0.78). In this way, our proposed multi-channel deep neural network for multi-modal medical imaging and clinical information for MTLE identification is outstanding from other single-channel algorithms.

### 4.2. Model Improvement

Among all misclassified cases, one epileptic patient was identified as the healthy control in every folder. After examination by radiologist, the patient showed no obvious abnormality on the T1 MRI image, while they showed bilateral metabolic asymmetry in the PET image. As shown in Figure 5, the T1 MRI image shows that the structure of the hippocampal is symmetrical and normal, but the metabolism on the right side is lower than that on the left.

To eliminate this kind of error, we added a neural network block before the entire network, identifying the asymmetries of a bilateral brain in PET images. The block is shown in Figure 6.

In this block, the normalized PET images were divided into two sides. Then, they were cut into cubes of 48 × 48 × 48 separately to calculate the differences of the symmetric cubes. The differences of the symmetric cubes were trained in a convolutional neural network to determine whether the pair of cubes belong to EP(1) or HC(0). 

This result must be added as an element to the typical clinical symptom and PDC vector, followed by trained structural T1 MRI, ^18^F-FDG PET/CT, clinic symptom, PDC data and the pre-trained PET result using the previous multi-channel 3D neural network in Figure 3. The only difference is that an element is added to the clinical-information channel whether PET is positive or not. The comparison result of MTLE identification before and after model improvement is shown in Table 2 and Figure 7. 

The MTLE identification with the PET pre-training block shows a better performance (Accuracy = 81.67% ± 10.33%, AUC = 0.80) than that of the previous experiment (Accuracy = 77.33% ± 14%, AUC = 0.78). Therefore, the improved integrated multi-channel 3D deep neural network we proposed in this paper achieved the best solution to the MTLE identification problem.

It is worth mentioning that the result shows a declined sensitivity, which is important to the diagnostic result. It should also be considered that the amount of MTLE data is small, and we divided the PET images into cubes of 48 × 48 × 48 and labeled them separately. These cubes may contain incomplete epileptic zones that destroy the integrity of biomarkers. This means images were identified as HC, which should have been identified as MTLE. The above reasons lead to a decrease in sensitivity.

## 5. Discussion

Epilepsy is a chronic brain disease with recurrent seizures and a prevalence of 1% [1]. With the development of medical technology, the diagnosis information can be obtained from different channels for the same disease. The causes of epilepsy are complex, and distinguishing different types of epilepsy accurately is difficult with a single mode of examination. Therefore, how to integrate multi-modal data for automated diagnosis of MTLE has become a challenge at present. In order to diagnosis MTLE from the age-matched health controls, we assess a fusion of multi-modal epilepsy medical information from structural T1 MRI, PET images, typical clinical symptoms and PDC by employing an integrated multi-channel 3D deep neural network with a PET pre-trained block. 

In a previous study, we used a single channel of MRI images to identify a relatively singular type of epilepsy, hippocampal sclerosis, with an accuracy of 81% and identified the MTLE with an accuracy of 78% [21]. Hence, it is possible that the PET image information and clinical symptom information may lead to an increase in identification accuracy. Zhang et al. used a deep learning framework for PET/MR image diagnosis in pediatric patients with TLE and achieved an accuracy of 90% [22]. We studied the idea of building a 3D CNN to process 3D image cubes in this paper. Even though our accuracy is lower than that in this experiment, it may be attributed to a broader classification of epilepsy and the high resolution of PET/MR images. Another study used a multi-channel deep neural network to integrate structural and functional MRI data and PDC information for TLE patients. The accuracy achieved was 70% according to a combination of different modals of medical data [23]. As can be seen from the comparison, the inclusion of PET data is one of the factors driving an accurate diagnosis of MTLE. Therefore, the experiment in our study demonstrated the potential of multi-channel CNN framework to integrate multi-modal medical data for MTLE identification and diagnosis. 

This study has several limitations. One limitation is only 15 MTLE cases were collected in this paper, so the data augmentation method was adopted to avoid over-fitting during training progress. This is due to the high cost of PET examination and patient privacy policies. In order to obtain accurate post-operative information as the gold standard, only 15 patients agreed to be included in the experiment. The MTLE data will continue to be collected in our subsequent studies. Another limitation is the lack of clinical information included in this experiment. We only took upper-abdominal abnormality and a feeling of fear into consideration. Actually, the appearance of external symptoms is related to brain area closely, which is affected by the transmission route of the abnormal discharge. Other symptoms and the order that the symptoms occur can also be biomarkers of MTLE diagnosis. This will be studied in our further work.

## 6. Conclusions

In order to diagnosis MTLE from the age-matched health controls, we assess a fusion of multi-modal epilepsy medical information from structural T1 MRI, PET images, typical clinical symptoms and PDC by employing an integrated multi-channel 3D deep neural network with a PET pre-trained block. Compared with single-modal inputs for single-channel deep neural network, our proposed method achieved a higher diagnosis accuracy of 81.67% ± 10.33%, while it performed well in all kinds of statistical indexes. The experiment result indicates that the proposed multi-channel deep neural network framework could diagnosis MTLE accurately and efficiently, which provides a feasible method for the application of multi-modal medical fusion in epilepsy diagnosis in the future.

## Figures and Tables

**Figure 1 bioengineering-10-01234-f001:**
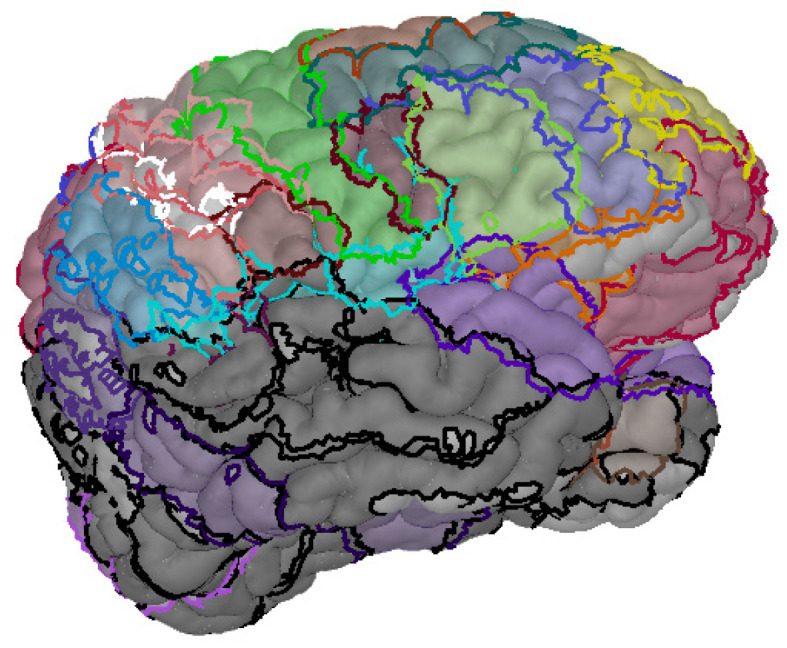
3D cerebral cortex model after pre-processing, registration and mapping to Broadmann template.

**Figure 2 bioengineering-10-01234-f002:**
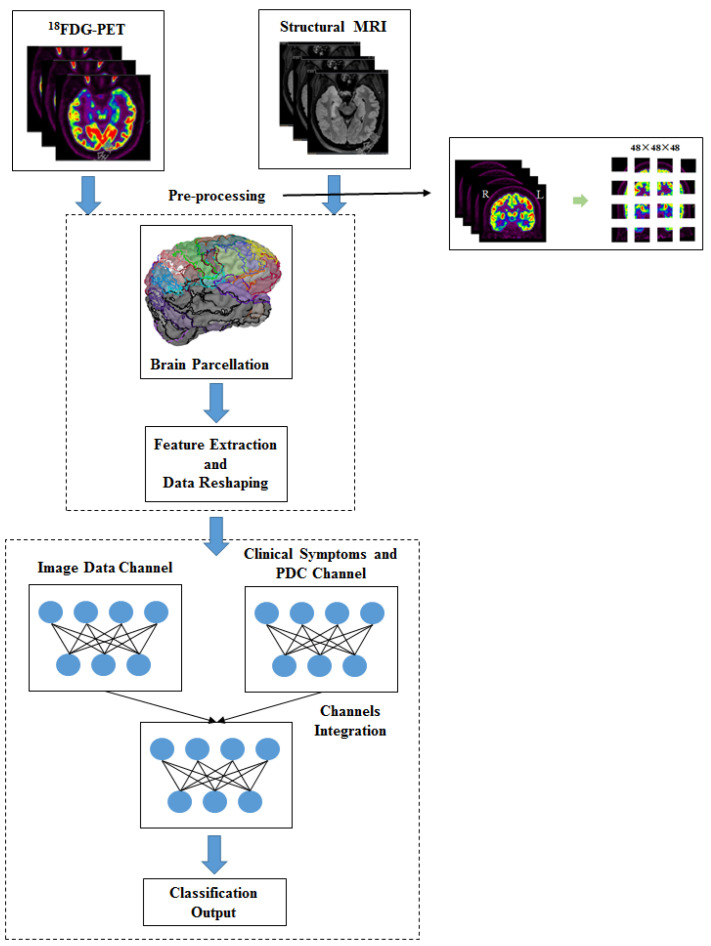
Experimental design schematic.

**Figure 3 bioengineering-10-01234-f003:**
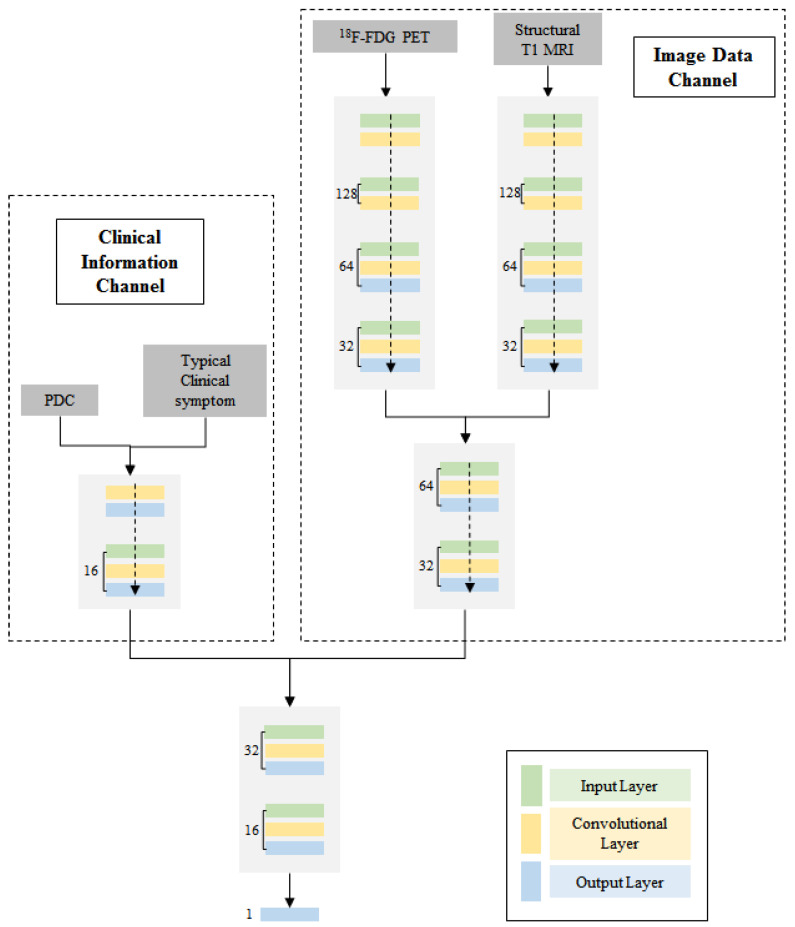
The proposed multi-channel deep CNN architecture.

**Figure 4 bioengineering-10-01234-f004:**
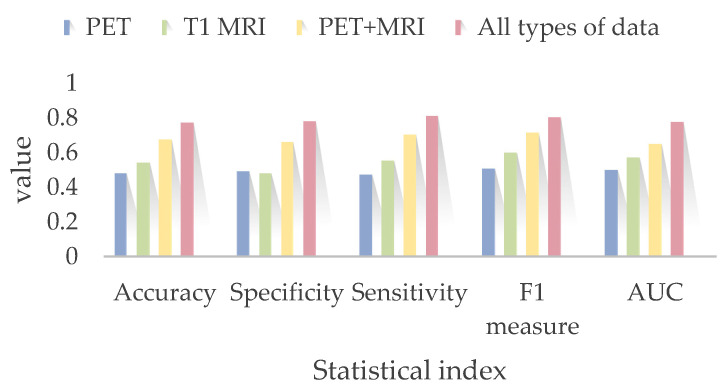
MTLE identification results using different types of inputs.

**Figure 5 bioengineering-10-01234-f005:**
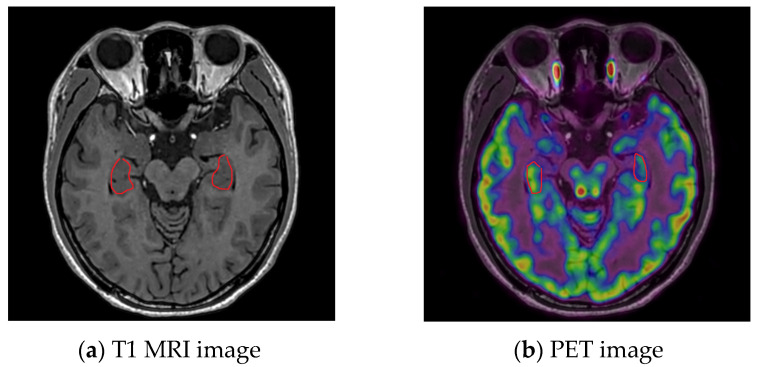
A misclassified patient with MRI(−) and PET(+) in hippocampal.

**Figure 6 bioengineering-10-01234-f006:**
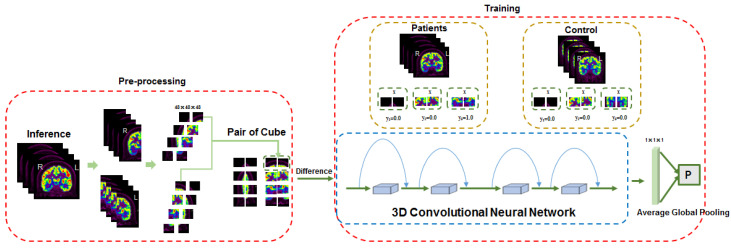
A convolutional neural network block to identify the asymmetries of a bilateral brain in PET images.

**Figure 7 bioengineering-10-01234-f007:**
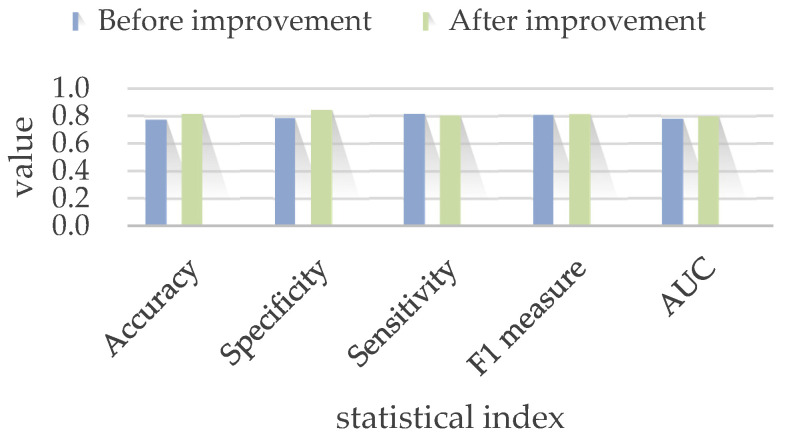
Comparison result of MTLE identification before and after model improvement.

**Table 1 bioengineering-10-01234-t001:** MTLE Identification Results using Different Types of Inputs.

Inputs	PET Images Along	Structural T1 MRI Along	PET Images and T1 MRI	All Information
Neural Network	Image-Data Channel	Image-Data Channel	Image-Data Channel	Whole Network
Accuracy	48% ± 19.85%	54.33% ± 7.67%	67.33% ± 50.25%	77.33% ± 14%
Specificity	49.33% ± 20.01%	48.35% ± 6.63%	66.35% ± 7.32%	78.44% ± 19.35%
Sensitivity	47.85% ± 15.73%	55.67% ± 3.38%	70.68% ± 19.34%	81.51% ± 15.86%
F1-measure	50.74% ± 13.21%	60.30% ± 11.59%	71.65% ± 11.94%	80.71% ± 2.21%
AUC	0.50	0.57	0.65	0.78

**Table 2 bioengineering-10-01234-t002:** Comparison result of MTLE identification before and after model improvement.

Model	Accuracy	Specificity	Sensitivity	F1 Measure	AUC
Before improvement	77.33% ± 14%	78.44% ± 19.35%	81.51% ± 15.86%	80.71% ± 2.21%	0.78
After improvement	81.67% ± 10.33%	84.33% ± 16.58%	80.33% ± 50.25%	81.33% ± 14%	0.80

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
