# Peer review of "An Integrated Multi-Channel Deep Neural Network for Mesial Temporal Lobe Epilepsy Identification Using Multi-Modal Medical Data"

_bioengineering, 2023, doi:10.3390/bioengineering10101234_

Round 1

Reviewer 1 Report

The topic of this study is certainly of interest and the proposed multichannel deep neural network framework could bring benefits to the diagnosis of epilepsy in the future. The statistical part is good and the figures are sufficiently informative. Despite this there are several points that should be reviewed:

-Line 16-17: I suggest to change the sentences “However, the … epilepsy accurately.” in: “The causes of epilepsy are complex and distinguishing different types of epilepsy accurately is challenging with a single mode of examination.”

-Line 17: use “aim” instead of “objective”

-Figure 1 is not clear, the color code does not have an immediate meaning as it lacks a legend, the resolution is low and the caption is not informative enough. 

-The sample size is small, especially for a study of this type where the aim is to make an accurate classification between medial temporal lobe epilepsy patients and healthy controls by adopting a multi-channel 3D neural network.

-How many channels did the MRI coil have?

-Was the preprocessing of the MR images done following references? If yes, indicate them.

-I’m surprised that there's a paragraph missing discussing the results. If the authors want to include the discussion in the results they must indicate it and certainly expand the text. Authors should highlight the importance of their findings and contextualize them with respect to the scientific context.

-Limitations of the study are missing

-Generally the entire text should be revised for better understanding by the reader.

Moderate editing of English language required

Reviewer 2 Report

The paper is interesting and the presentation is acceptable but need minor modifications before publication. the comments are prepared below:

-Use this sentence in the abstract is inappropriate ‘’However, the causes of epilepsy are complex.’’

 -In the Abstract, please avoid using words such as, thus, Hence, therefore. Because abstract is not an appropriate section to discuss and make conclusion.

-In the section keywords, please introduce ‘’MTLE’’ completely.

Minor English polishing is required.

Author Response

Dear professor,

Thank you for your suggestions!

1.-Use this sentence in the abstract is inappropriate ‘’However, the causes of epilepsy are complex.’

Thank you for your suggestions. I modified to “However, the causes of epilepsy are complex and distinguishing different types of epilepsy accurately is challenging with a single mode of examination.”

2. -In the Abstract, please avoid using words such as, thus, Hence, therefore.  Because abstract is not an appropriate section to discuss and make conclusion.

Thank you for the professional suggestion. I delete the exact result and conclusion in the abstract.

3. -In the section keywords, please introduce ‘’MTLE’’ completely.

I modified the “MTLE” to “Mesial temporal lobe epilepsy” in the section keywords.

Reviewer 3 Report

Healthy contol is not healthy.

After improvement, specificity was improved, however, no improvement of sensitivity. Author should discuss these results.

OK

Author Response

Dear professor,

Thank you for your suggestions!

1.-Healthy control is not healthy.

Actually it’s right. The health control is not healthy but healthy in brain. In the section 2.1, we mentioned that “It is worth mentioning that, these 15 health controls are patients who had disease that requires a PET/CT examination but normal in brain. It is because there is nuclear radiation in PET/CT examination and it is harmful to healthy people to undergo this experiment.”

2. -After improvement, specificity was improved, however, no improvement of sensitivity. Author should discuss these results.

Considered that the amount of MTLE data is small, we divided the PET images into cubes of 48×48×48 and labeled them separately. These cubes may contain incomplete epileptic zone that destroy the integrity of biomarkers. This allows images were identified to HC, which should have been identified as MTLE. The above reasons lead to a decrease in sensitivity.

Round 2

Reviewer 1 Report

The authors corrected their manuscript following the reviewers' recommendations. The article can be accepted in its current form.

Reviewer 3 Report

I found significant improvement of the manuscript.